# DIFFUSION MINIMIZATION AND SHEAF NEURAL NETWORKS FOR RECOMMENDER SYSTEMS

## ABSTRACT

Graph Neural Networks (GNN) are well-known for successful applications in recommender systems. Despite recent advances in GNN development, various authors report that in certain cases GNN suffer from so-called oversmoothing problems. Sheaf Neural Networks (SNN) is one of the ways to address the issue of oversmoothing. In the present work we propose a novel approach for training SNN together with user and item embeddings. In that approach parameters of the sheaf are inferred via minimization of the classical BPR loss and sheaf diffusion on graphs subjected to orthogonality and consistency constraints. Performance of the novel technique is evaluated on synthetic test cases and standard benchmarks for recommendations.

## 1 INTRODUCTION

Graph Neural Networks (GNN) Gao et al. (2022) is a rapidly developing research area in Machine-Learning. GNN has been already successfully applied to solve problems in different areas including Recommender Systems He et al. (2020), drug discovery Duvenaud et al. (2015); Han et al. (2021) search for new materials Park & Wolverton (2020), materials properties prediction Louis et al. (2020) and Natural Language Processing Gui et al. (2019). One of the possible reasons for such success is utilization of information about pairwise interactions due to processing of graph connections data.

Despite numerous successful applications, there is a space for further improvements. One of the issues that reduces GNN performance is so-called oversmoothing effect Rusch et al. (2023). There is a variety of quantitative definitions of oversmoothing Chen et al. (2020a), but all of them are related to the variability of vertices feature vectors over the graph. In the extreme case of high oversmoothing feature vectors associated with each vertex of graph are almost constant. The latter means that such features are not informative and can not be utilized to solve machine-learning problems.

The oversmoothing occurs because of a message-passing procedure that allows one to aggregate information from neighbours. It has been reported Rusch et al. (2023) that smoothing of feature vectors can increase the quality of machine-learning model trained on the graph, because it performs denoising of vertices feature vectors. Therefore, addressing the issues of oversmoothing is not so straightforward.

Different techniques emerged to tackle the oversmooting in GNN. One of the options of oversmoothing reduction is regularization of GNN training. In Zhou et al. (2021) regularization technique that promotes variability of graph vertices features was proposed. Various stochastic methods for graph edges dropout can reduce the oversmoothing effect ?Hasanzadeh et al. (2020). In other works authors consider modifications of message-passing procedure to produce informative features. Such methods can be inspired by physical systems on graphs Rusch et al. (2022a); Eliasof et al. (2021); Wang et al. (2023); Giovanni et al. (2022). Alternative methods for modification of message-passing procedure include Gradient Gating Rusch et al. (2022b). It has been demonstrated that addition of residual connections to Graph Neural Networks can He et al. (2016); Li et al. (2019); Chen et al. (2020b).

In recent years a novel technique for oversmoothing tackling emerged. The novel approach is based on Graph Sheaf Theory Curry (2014) and is referred to as Sheaf Neural Networks (SNN) . In SNN each graph vertice and edge is equipped with a linear space and linear maps between vector spaces

of edges and vertices connected by those edges. Such construction provides more accurate description of data and relations between graph vertices Hansen & Gebhart (2020). The latter results in improvement of recommender systems Barbero et al. (2022).

SNNs can be considered as a modification of the message-passing algorithm in classical GNNs, where feature vectors from neighbours are passed directly. In SNN parameters feature vectors of neighbouring vertices are multiplied by matrices that are computed for each pair of vertices. For instance, if one wants to pass vector $\boldsymbol{x}$ from vertex $u$ to verix $v$, then $\boldsymbol{x}$ should be mapped first to the linear space of edge $uv$: $\boldsymbol{y} = \boldsymbol{A}_u \boldsymbol{x}$ and mapped to the space of features at $v$: $\boldsymbol{x}^* = \boldsymbol{A}_v^\top \boldsymbol{A}_u \boldsymbol{x}$. The procedure description has certain important benefits. First of all, it provides means for oversmoothing reduction by aligning vectors from different vertices in modified message-passing algorithm, Secondly, it provides a consistent way for comparison between vectors associated with different vertices by mapping such vectors to the space of the edge that connects these vertices Purificato et al. (2024). This is beneficial for classical recommender methods based on dot-product of user and item embeddings.

In the current work we present a novel approach on learning SNN parameters. Basically, we introduce orthogonality constraint on sheaf linear maps similar to Barbero et al. (2022) and the novel consistency constraint explained in the Section 2. Sheaf linear maps are parameterized as functions of feature vectors and are tuned to minimize the vertex feature vector diffusion. Finally, Bayesian Personalized Ranking loss-function is added to train the recommender. Cases of known and learnable vertices feature vectors are considered. Our work adopts the approach similar to Bamberger et al. (2024). However, we utilize a different regularisation approach to learn the sheaf structure.

## 2 METHODOLOGY

In the present section we provide Mathematical formulation of the approach for SNN training. First we give a brief introduction to message passing on graphs equipped with a sheaf structure in the subsection 2.1. Secondly, we describe the constraints on sheaf linear transformations in the subsection 2.2. Finally, the loss function for recommender training is provided in subsections 2.3. We conclude the section by providing formulations of some Theorems that are discussed in the Appendix.

### 2.1 SHEAF THEORY AND MESSAGE PASSING

Sheaf theory for graphs is a huge subject in Modern Mathematics with a variety of concepts of different complexity. However, in our work we utilized only one of them to develop the novel approach for SNN training. The most essential concept for our work is related to linear spaces over each vertice and edge of the graph.

If $V$ is the set of vertices and $E$ is the set of edges of the graph $\mathcal{G}$, then the sheaf structure implies that there is a vector space $L(v)$, associated with each $v \in V$ and a vector space $L(e)$, associated with each $e \in E$. Moreover, for any edge $e$ that connects vertices $u$ and $v$ there is a linear map:

$$\boldsymbol{A}(v) : L(v) \to L(e) \tag{1}$$

Therefore, one can pass the feature vector $\boldsymbol{x}(v)$ from vertex $v$ to vertex $u$ by combining the linear maps in the equation 1:

$$\boldsymbol{A}^\top(u)\boldsymbol{A}(v) : L(v) \to L(u) \tag{2}$$

These maps can be utilized for message-passing procedures. If $D(v)$ - is the degree of the vertex $v$, then the message passing can be defined as follows:

$$M_u(x) = \sum_{v:(vu)\in E} \frac{1}{D(u)} \boldsymbol{A}^\top(u)\boldsymbol{A}(v)\boldsymbol{x}(v) \tag{3}$$

Here $M_u(x)$ is the result of the message-passing of feature vectors $\boldsymbol{x}$ for the vertex $u$. In other words, the input for $M$ is the vector field one the graph and a result is the vector field on the graph, which can be evaluated at each graph vertex. The summation in the equation 3 is performed over all vertices $v$ such that $u$ and $v$ are connected by the edge: $(uv) \in E$. Therefore, message-passing can be defined for weighted graphs naturally:

$$M_u(x) = \sum_{v} \frac{w(u,v)}{\sum_{v^*} w(u,v^*)} \boldsymbol{A}^\top(u)\boldsymbol{A}(v)\boldsymbol{x}(v) \tag{4}$$

Where $w(v, u)$ is a weight of the edge that connects $u$ and $v$. Accordingly, the case of the weighted fully-connected graph can be considered without loss of generality. In the case of weighted graph, we denote as $D(u)$ the normalization coefficient:

$$D(u) = \sum_v w(v, u) \tag{5}$$

Operations in equation 4 can be rearranged as following:

$$M_u(x) = \frac{1}{D(u)} \boldsymbol{A}^\top(u) \sum_v w(u, v) \boldsymbol{A}(v) \boldsymbol{x}(v) = \boldsymbol{A}^\top(u) \left( \sum_v \frac{w(u, v)}{D(u)} \boldsymbol{y}(v) \right) \tag{6}$$

Here the following vector is defined: $\boldsymbol{y}(v) = \boldsymbol{A}(v)\boldsymbol{x}(v)$. In other words, $\boldsymbol{y}(v)$ is simply the result of mapping a feature vector $\boldsymbol{x}(v)$ related to the vertex $v$ to the particular vertex $u$. As a result, in the case of SNN the message-passing and feature aggregation happens in the edge spaces. The result of aggregation is then mapped back to the linear spaces, associated with each vertex.

## 2.2 Constraints

In the present work we utilize several constraints on sheaf linear maps. The first one is the orthogonality constraint similar to Barbero et al. (2022). We assume that the dimension of the edge space is (significantly) lower than the dimension of the linear space associated with the vertex. Therefore, orthogonality constraint takes the following form:

$$\boldsymbol{A}(u)\boldsymbol{A}^\top(u) - \boldsymbol{I} = 0 \tag{7}$$

Here $\boldsymbol{I}$ is the identity matrix. In other words, $\boldsymbol{A}(u)$ projects vector $\boldsymbol{x}(u)$ and performs some rotations in the projected space to compute the edge feature vector.

There is a geometric interpretation of the orthogonality constraint Barbero et al. (2022). Graph Sheaf can be considered as a discretisation of a vector bundle over the manifold. In the case of the tangent bundle over the Riemannian Manifold so-called parallel transport can be introduced. In other words, it is possible to take a tangent vector at one point on the manifold and bring it along a curve to the other point. In such a procedure the length of the vector does not change and the smooth curve that connects two points on the manifold determines the linear map between tangent spaces at these points that preserves the vector length. Therefore, such a linear map is a rotation, which is expressed in the equation 7.

The second constraint we impose is consistency requirement. We suppose that linear mappings depend only on the feature vector:

$$\boldsymbol{A}(u) = \boldsymbol{A}\big(\boldsymbol{x}(u)\big) \tag{8}$$

It follows immediately from equation 7 that $\boldsymbol{A}^\top(u)\boldsymbol{A}(u)$ is a projection operator. One can think about this linear map $\boldsymbol{A}^\top(u)\boldsymbol{A}(u) : L(u) \to L(u)$ as a feature denoising. Therefore, it is reasonable to suppose that sheaf linear map $\boldsymbol{A}(x(u))$ should not depend on the noisy component of feature vector $\boldsymbol{x}(u)$. The latter can be formulated as a constraint:

$$\boldsymbol{A}\big(\boldsymbol{x}(u)\big) = \boldsymbol{A}\Big( \boldsymbol{A}^\top\big(\boldsymbol{x}(u)\big) \boldsymbol{A}\big(\boldsymbol{x}(u)\big) \boldsymbol{x}(u) \Big) \tag{9}$$

For the purpose of simplicity, we introduce the notation for the linear operator:

$$\boldsymbol{P}(u) = \boldsymbol{A}^\top(u)\boldsymbol{A}(u) = \boldsymbol{A}^\top\big(\boldsymbol{x}(u)\big)\boldsymbol{A}\big(\boldsymbol{x}(u)\big) \tag{10}$$

## 2.3 Sheaf Learning

In principle, a single sheaf on the graph can be trained independently on the ultimate goals of the recommender system. We doing that by minimizing sheaf diffusion under orthogonality and consistency constraints. The latter is performed via appropriate weighting of loss functions for constraints and for the target metric.

The loss for orthogonality constraint can derived simply from the equation 7:

$$\mathcal{L}_{\text{orth}} = \sum_{i=1}^{N} \frac{1}{N} \text{trace}\Big( \big(\boldsymbol{A}(u_i)\boldsymbol{A}^\top(u_i) - \boldsymbol{I}\big)^\top \big(\boldsymbol{A}(u_i)\boldsymbol{A}^\top(u_i) - \boldsymbol{I}\big) \Big) \tag{11}$$

Here $N$ is the batch size.

The loss function for the consistency constraint can be computed in a similar fashion:

$$\mathcal{L}_{\text{cons}} = \sum_{i=1}^{N} \frac{1}{N} \boldsymbol{x}^{\top}(u_i)\big(\boldsymbol{A}(u_i) - \boldsymbol{A}(u_i)\boldsymbol{P}(u_i)\big)^{\top}\big(\boldsymbol{A}(u_i) - \boldsymbol{A}(u_i)\boldsymbol{P}(u_i)\big)\boldsymbol{x}(u_i) \qquad (12)$$

We minimizing sheaf diffusion by forcing feature vectors not to change in the message-passing procedure:

$$\mathcal{L}_{\text{diff}} = \sum_{i=1}^{N} \frac{1}{N} \big(M_{u_i}(\boldsymbol{x}) - \boldsymbol{x}(u_i)\big)^{\top}\big(M_{u_i}(\boldsymbol{x}) - \boldsymbol{x}(u_i)\big) \qquad (13)$$

Loss functions introduced in equation 11, equation 12 and equation 13 are combined into a single loss-function:

$$\mathcal{L}_{\text{comb}} = \text{sg}(w_{\text{orth}})\mathcal{L}_{\text{orth}} + \text{sg}(w_{\text{cons}})\mathcal{L}_{\text{cons}} + \text{sg}(w_{\text{diff}})\mathcal{L}_{\text{diff}} \qquad (14)$$

Here sg - stop gradient operation. In other words, derivatives of loss-function weights are not computed in the back-propagation step.

Me adopt the approach similar to bareer-function method that is well-known in optimization. In other words, weights $w_{\text{orth}}, w_{\text{cons}}, w_{\text{diff}}, w_{\text{bprl}}$ are proportional to:

$$\begin{cases} w_{\text{orth}} \propto 1, \\ w_{\text{cons}} \propto \exp\Big(-\kappa_{\text{cons}}\sqrt{N}\mathcal{L}_{\text{orth}}\Big), \\ w_{\text{diff}} \propto \exp\Big(-\kappa_{\text{diff}}\sqrt{N}\max\big(\mathcal{L}_{\text{orth}}, \mathcal{L}_{\text{cons}}\big)\Big) \end{cases} \qquad (15)$$

Here $\kappa_{\text{cons}}, \kappa_{\text{diff}}$ are tunable hyperparameters.

The following normalization approach is utilized:

$$w_{\text{orth}} + w_{\text{cons}} + w_{\text{diff}} = 1 \qquad (16)$$

In the case of the weighting procedure as in equation 15 and equation 21 loss-functions are minimized sequentially. It is simple to see that if the orthogonality constraint is not satisfied, all other weights are close to zero. Since orthogonality error follows below a certain level, consistency error starts to reduce and so on. Finally, diffusion and BPR loss are minimized under orthogonality and consistency constraints. The factor $\sqrt{N}$ is introduced to reduce the hyper-parameters dependency on a batch-size.

## 2.4 SHEAVES FOR REGRESSION AND LINK PREDICTION

Sheaves on graphs can be tuned to solve various machine-learning problems on graphs. However, two important steps are required.

The first one is related to message passing. It is common to utilize information from vertices that are within the distance of a given number of hops from a given vertex. We have described only one-hop approach in the equation 3. It is simple to see that the combination of several convolutions like in equation 3 is equivalent to exploration of graph in depth. In our numerical experiments we usually use three convolutions.

The second one addresses the issue of learning objectives. If we consider graph regression problems or link prediction, then we normally have a loss-function $\mathcal{L}_{\text{target}}$ such as a Mean-Squared Error (MSE) or BPR-loss. This target loss function is introduced similarly to the equation 14

$$\mathcal{L}_{\text{comb}}^{*} = \text{sg}(w_{\text{orth}})\mathcal{L}_{\text{orth}} + \text{sg}(w_{\text{cons}})\mathcal{L}_{\text{cons}} + \text{sg}(w_{\text{diff}})\mathcal{L}_{\text{diff}} + \text{sg}(w_{\text{target}})\mathcal{L}_{\text{target}} \qquad (17)$$

The weight for the learning-objective or target is $w_{\text{target}}$. We utlize the same technique as in the equation 15 to compute weights:

$$
\begin{cases}
w_{\text{orth}} \propto 1, \\
w_{\text{cons}} \propto \exp\left(-\kappa_{\text{cons}}\sqrt{N}\mathcal{L}_{\text{orth}}\right), \\
w_{\text{diff}} \propto \exp\left(-\kappa_{\text{diff}}\sqrt{N}\max\left(\mathcal{L}_{\text{orth}}, \mathcal{L}_{\text{cons}}\right)\right) \\
w_{\text{target}} \propto \exp\left(-\kappa_{\text{diff}}\sqrt{N}\max\left(\mathcal{L}_{\text{orth}}, \mathcal{L}_{\text{cons}}, \mathcal{L}_{\text{diff}}\right)\right)
\end{cases}
\tag{18}
$$

With the normalization constraint:

$$
w_{\text{orth}} + w_{\text{cons}} + w_{\text{diff}} + w_{\text{target}} = 1 \tag{19}
$$

Sometime it is more efficient to utilize alternative hierarchy:

$$
\begin{cases}
w_{\text{orth}} \propto 1, \\
w_{\text{cons}} \propto \exp\left(-\kappa_{\text{cons}}\sqrt{N}\mathcal{L}_{\text{orth}}\right), \\
w_{\text{target}} \propto \exp\left(-\kappa_{\text{target}}\sqrt{N}\max\left(\mathcal{L}_{\text{orth}}, \mathcal{L}_{\text{cons}}\right)\right) \\
w_{\text{diff}} \propto \exp\left(-\kappa_{\text{diff}}\sqrt{N}\max\left(\mathcal{L}_{\text{orth}}, \mathcal{L}_{\text{cons}}, \mathcal{L}_{\text{target}}\right)\right)
\end{cases}
\tag{20}
$$

With the normalization constraint:

$$
w_{\text{orth}} + w_{\text{cons}} + w_{\text{diff}} + w_{\text{target}} = 1 \tag{21}
$$

Finally, it is worth noting that in the case of classical regression or link-prediction problems we do not have feature vectors associated with vertices or edges. In our work we learn those features to minimize the loss function in the equation 17.

Eventually, it is worth noting that we assume that sheaf linear maps $A(u)$ and projection operators $P(u)$ depend only on the vertex feature vector. Therefore, feature vectors provide accurate users and item descriptions. As a result, there is a potential to utilize the presented approach to address the cold-start issue.

## 2.5 CONNECTION WITH THE GRAPH LAPLACIAN

In our approach for GNN linear operations are utilised in the message passing as in the equation 3 or equation 4. Due to the linearity of the operations in the message passing, this procedure can be decomposed into three steps: mapping of vertices feature vectors to a certain ambient space $x \to y$, simple message passing, where features vectors are simply averaged over the neighbors, pseudo-inverse linear map $y \to x$. Therefore, the theoretical analysis is simple to conduct in the space of $y$ vectors. In this case the message passing can be performed as follows:

$$
\hat{\boldsymbol{y}}(u) = \sum_{v \in V} \frac{w(u, v)}{D(u)} \boldsymbol{y}(v) \tag{22}
$$

Here $\hat{\boldsymbol{y}}(u)$ is the feature vector at the vertex $u$. This equation can be written in the matrix form:

$$
\hat{\boldsymbol{y}}_\alpha = \boldsymbol{D}^{-1}\boldsymbol{W}y_\alpha \tag{23}
$$

Here $\alpha$ is the coordinate index of the feature vector $y$, $W$ is the matrix of weights: $W_{uv} = w(u, v)$ and $\boldsymbol{D}$ is the diagonal matrix with elements $D_u = \sum_v W_{uv}$. The difference between $y$ and $\hat{\boldsymbol{y}}$ is simply:

$$
\hat{y}_\alpha - \boldsymbol{y}_\alpha = (\boldsymbol{I} - \boldsymbol{D}^{-1}\boldsymbol{W})y_\alpha \tag{24}
$$

Here $I$ is the identity matrix. The matrix $\boldsymbol{L}_{rw} = \boldsymbol{I} - \boldsymbol{D}^{-1}\boldsymbol{W}$ is a normalized matrix of the Graph-Laplacian operator. It is simple to show that the matrix $L_{rw}$ has a basis of eigen-vectors and all

eigen-values are in the interval $[0; 1]$. Given that fact, vector $y(u)$ can be decomposed as a linear combination of eigen-functions $\phi_1(u), ..., \phi_N(u)$ on graph with $N$ vertices:

$$y_\alpha(u) = \sum_{a=1}^{N} \eta_{\alpha a} \phi_a(u) \tag{25}$$

Here $\eta_{\alpha a}$ is a matrix of coefficients. The decomposition in the equation 25 provides bounds on the diffusion process that are formulated in the Theorem 2.1:

**Theorem 2.1.** *Suppose that vector-valued function $y(u)$ on a (weighted) graph with the set of vertices $V$ with cardinality $N$ is decomposed as:*

$$y_\alpha(u) = \sum_{a=1}^{N} \eta_{\alpha a} \phi_a(u)$$

*and that eigen-values of eigen-functions in the decomposition above are in the interval $[\lambda_{min}, \lambda_{max}] \subset [0; 1]$, then the there exist such norm $|| \cdot ||$ in the space of vector-valued functions on $V$ such that the norm of $(I - D^{-1})W)y$ is bounded as follows:*

$$\lambda_{min} ||\boldsymbol{y}|| \leq ||\boldsymbol{I} - \boldsymbol{D}^{-1}\boldsymbol{W})\boldsymbol{y}|| \leq \lambda_{max} ||\boldsymbol{y}||$$

The immediate consequence of the Theorem 2.1 is that there is no diffusion, if only eigen-functions with zero eigen-values have non-zeros coefficients in the equation **??**. Such eigen-functions are constant on each connected component of the graph $G$. Therefore, nontrivial representative features $y(u)$ lead to non-zero diffusion in the message-passing. Moreover, it is simple to show that if there is no learning objective, then there exist a sheaf with zero diffusion. More precisely, the following statement holds:

**Theorem 2.2.** *Suppose that the vector-valued function $x(u)$ is defined on a graph $G = (V, E)$. If for any vertex $u \in V$ the Euclidean norm $||x(u)||_2 = 1$, then there exist linear transformations $A(u)$ that satisfy orthogonality and consistency constraints and have zero diffusion.*

The immediate consequence of the Theorem 2.2 is that it is quite simple to construct a sheaf with zero diffusion. The feature vector $y(u)$ is constant in this case. Therefore, such features are note in a good fit to machine-learning tasks.

Apart from providing the example of sheaves with zero diffusion, the Theorem 2.2 explains the importance of the hierarchy in the loss function. For example, if the target loss-function (aka cross-entropy) is the latest in the hierarchy, then for certain values of hyperparameters the constant sheaf with zero diffusion is be learnt. Therefore, the target loss-function can be hardly minimised because of the constant input.

In order to address the issues of trivial input features for machine-learning problem, one can assign the lowest priority to the diffusion minimisation. In this case target loss is minimised and sheaf diffusion minimisation provides Tikhonov's regularisation:

**Theorem 2.3.** *Suppose that for a given (weighted) graph $G = (V, E)$, coordinates of a vector-valued function $x(u)$ span the space that contains $\phi_1, ..., \phi_m$. Then for a given scalar function $f : V \to \mathbb{R}$ the RMSE error of the solution for hierarchical loss minimization is bounded by the $\ell_2$ norm of the projection of $f$ on the space spanned by $\phi_{m+1}, ..., \phi_N$.*

The Theorem 2.3 states that the error of the solution for the regression problem can be made arbitrarily small if the dimension of vertex and edge feature vectors is high enough.

It is important to notice that in our numerical experiments we utilise the hierarchy of loss-functions that should provide trivial feature vector $y(u)$ according to the Theorem **??**. In the numerical experiments graph embedding is learnable and it is initialized randomly. Therefore, assumptions of the theorem do not hold, because feature vectors change through the training process. In the case of learnable features, minimisation of the diffusion provides regularisation of noisy feature vectors.

## 3 NUMERICAL EXAMPLES

We validate the novel approach for SNN training on two groups of test-cases. The first group consists of synthetic examples with simple fully connected weighted graph and simple vertex features.

In the second group we compare our approach with other GNN methods on benchmarks in recommendations.

## 3.1 SYNTHETIC DATA

In this example we consider two options for graph generation: uniform sampling of vertices from the unit cube in $\mathbb{R}^3$. Edges weights are assigned via Radial Basis Functions (RBF) as follows:

$$w(v, u) = \exp\left(-\gamma|\xi(v) - \xi(u)|^2\right) \tag{26}$$

Here $\xi(u)$ and $\xi(v)$ are coordinate vectors of vertices $u$ and $v$ respectively, and $\gamma$ is a parameter.

First $m$ functions $\phi_1, ..., \phi_m$ of Laplacian operator Belkin & Niyogi (2008) and $n$-dimesnional feature vectors are generated via random $n \times m$ matrix $\chi$:

$$x_i(u) = \sum_{j=1}^{m} \chi_{ij} \phi_j(u) \tag{27}$$

Here $x(u)$ is a feature vector associated with the vertex $u$ and $i = 1, ..., n$ is a coordinate index. In addition to that we generate random function $f$ in the form:

$$f = \sum_{i=1}^{m} q_i \phi_i \tag{28}$$

The objective is to learn the sheaf and approximate $f$. In other words, we use sheaves to solve for the regression problem:

$$f(u) \approx \sum_{i=1}^{m} y_i(u) \tag{29}$$

We consider two cases of hierarchical loss: orthogonality, consistency, diffusion and MSE vs orthogonality, consistency, MSE and diffusion.

In the present test case we demonstrate that the sheaf can be trained on the data to achieve almost zero diffusion:

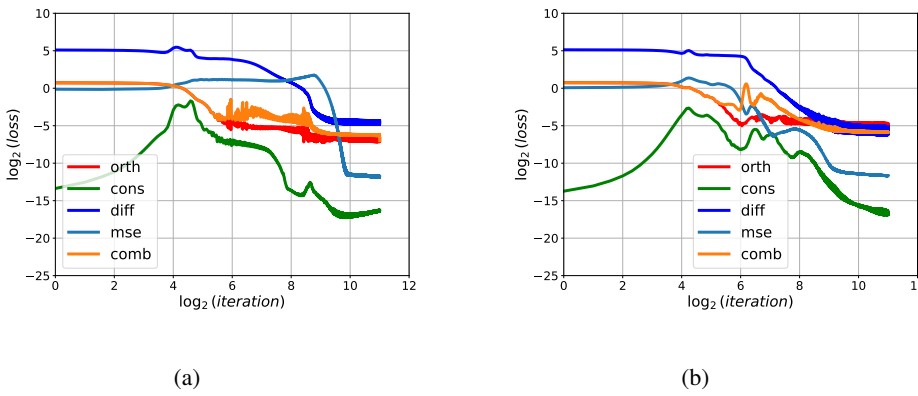

(a)  (b)

Figure 1: Plot of loss functions for different cases of loss-functions hierarchy: orthogonality, consistency, diffusion and MSE (left) vs orthogonality, consistency, MSE and diffusion (right).

This experiment shows that hierarchy of loss functions is important for convergence. In this particular example both training strategies are fine. However, convergence is faster in the case, when the MSE is not the last in the hierarchy. This is typical for clean features. It has been found empirically that in the case of noisy data it is better to minimize the target loss-function (aka MSE or BPR) in the last place.

## 3.2 RECOMMENDER SYSTEMS

In order to evaluate the benefits of SNN in recommender systems we compare it against LightGCN He et al. (2020) and UltraGCN Mao et al. (2021) on benchmark datasets: MovieLens Xiong (2021), Facebook Shapira et al. (2013) and Yahoo Yahoo!.

We consider SNN with three sheaf layers. The first layer is trained to with all three loss functions: $\mathcal{L}_{\text{orth}}$, $\mathcal{L}_{\text{cons}}$, $\mathcal{L}_{\text{diff}}$. All other sheaf layers consider only the value of BPR in training process. Results of numerical experiments are summarized in the Tab. 1.

Table 1: Comparision with Benchmarks

| Method | P@10 | R@10 | NDCG@10 | P@20 | R@20 | NDCG@20 |
|---|---|---|---|---|---|---|
| MovieLens | | | | | | |
| SheafGCN | 0.139 | 0.139 | 0.305 | 0.110 | 0.220 | 0.450 |
| UltraGCN | 0.170 | 0.072 | 0.240 | 0.140 | 0.120 | 0.260 |
| LightGCN | 0.002 | 0.001 | 0.004 | 0.004 | 0.003 | 0.005 |
| Facebook | | | | | | |
| SheafGCN | 0.009 | 0.050 | 0.071 | 0.007 | 0.090 | 0.120 |
| UltraGCN | 0.020 | 0.060 | 0.066 | 0.011 | 0.090 | 0.060 |
| LightGCN | 0.001 | 0.002 | 0.003 | 0.001 | 0.004 | 0.003 |
| Yahoo | | | | | | |
| SheafGCN | 0.028 | 0.012 | 0.138 | 0.041 | 0.021 | 0.162 |
| UltraGCN | 0.045 | 0.143 | 0.101 | 0.019 | 0.079 | 0.117 |
| LightGCN | 0.001 | 0.003 | 0.120 | 0.001 | 0.014 | 0.070 |

We utilize three metrics: precision@k (P@k), recall@k (R@k), and NDCG@k. Where:

1. Precision@K (P@K): measures the proportion of the recommended items that are relevant to the user among the top K items.

2. Recall@K (R@K): measures the proportion of relevant items that the system successfully recommended among the top K items.

3. NDCG@K, or Normalized Discounted Cumulative Gain at rank K: used to assess the usefulness of a ranking system. It does this by considering the relevance of the items in the ranked list and their positions in the list

Comparison of SNN with classical GCN demonstrates that conventional methods provide a more relevant list of candidates for recommendation. However, the ranking within the list of candidates performed by SNN is more accurate in comparison with GCN.

It is important to note the significant role of orthogonality and consistency constraints in the learning process. The latter is illustrated by the ablation study performed on the Facebook dataset and summarized in the table below:

Table 2: Ablation Study

| Regularisation | $w_{\text{orth}} = w_{\text{cons}} = 0$ | $w_{\text{orth}} \neq 0$ | $w_{\text{cons}} \neq 0$ | $w_{\text{orth}}, w_{\text{cons}} \neq 0$ |
|---|---|---|---|---|
| NDCG@50 | 0.0998 | 0.3490 | 0.0299 | 0.3504 |

## 4 CONCLUSION

The SNN is an attractive method for recommender systems from both theoretical and practical points of view. In SNNs, feature vectors of users and items are mapped to a single vector space by sheaf linear maps. The latter enables consistent comparison of user and item vectors, thereby simplifying the theoretical analysis of recommendations through SNNs.

The practical benefits of SNNs include feature denoising and reduction of the oversmoothing as illustrated by numerical experiments with synthetic and real data. Moreover, SNN achieves similar recommendation quality in comparison with classical GCN methods. Therefore, further development of SNNs is a promising area of research.

In summary, one of the main contributions of the present work is the novel approach for SNN training via loss-function minimization under constraints. Constraints introduced provide sheaf linear maps regularization and simplify the inference procedure: SNN requires only feature vectors to compute sheaf linear maps. In addition to that, we demonstrate that sequential application of sheaf layers can provide relevant recommendations for users. One of the limitations of the work is increased computational cost in comparison with GCN of similar architecture due to the need for sheaf linear maps calculation. Despite that fact, presented results demonstrate that SNN has high potential for applications in recommender systems.

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

# A APPENDIX

In the present section we provide proofs of the theorem from the subsection 2.5.

## A.1 PROOF OF THE THEOREM 2.1

In this subsection we introduce the norm in the space of vector-valued functions on graph so that lower and upper bounds for diffusion on the graph can be derived.

If $W$ is the matrix of edge weights for the graph with set of vertices $V$, then diagonal matrix $D$ can be introduced:

$$D(u, u) = \sum_{v \in V} W(u, v)$$

We assume that the matrix of weights is symmetric providing the symmetry of $D^{-1/2} W D^{-1/2}$. Matrix $D^{-1/2} W D^{-1/2}$ has non-negative eigen-values and basis of orthogonal eigen vectors $\psi_1, ..., \psi_N$, where $N$ is the number of vertices in $V$.

It is simple to check that eigen-vectors of $D^{-1/2} W D^{-1/2}$ are related to the eigen-vectors of $D^{-1} W$ by simple linear transformation:

$$\phi_a = D^{-1/2} \psi_a$$

Vector function $y : V \mapsto \mathbf{R}^m$ is represented as a linear combination:

$$y_\alpha = \sum_{a=1}^{N} \eta_{\alpha a} \phi_a = \sum_{a=1}^{N} \eta_{\alpha a} \boldsymbol{D}^{-1/2} \psi_a$$

Therefore, the following norm can be introduced:

$$\|\boldsymbol{y}\|_D^2 = \sum_{\alpha=1}^{m} y_\alpha^\top \boldsymbol{D} y_\alpha$$

It is simple to show that the square of the norm introduced is simply the square of the Frobenius norm of the matrix $\eta$

$$\|\boldsymbol{y}\|_D^2 = \sum_{\alpha=1}^{m} y_\alpha^\top \boldsymbol{D} y_\alpha = \sum_{\alpha=1}^{m} \sum_{a_1, a_2=1}^{N} \eta_{\alpha a_1} \eta_{\alpha a_2} \psi_{a_2}^\top \boldsymbol{D}^{-1/2} \boldsymbol{D} \boldsymbol{D}^{-1/2} \psi_{a_1} = \sum_{\alpha=1}^{m} \sum_{a=1}^{N} \eta_{\alpha a}^2$$

Therefore, operator $I - D^{-1}W$ acts on the vector field as following:

$$\hat{y}_\alpha - y_\alpha = (\boldsymbol{I} - \boldsymbol{D}^{-1}\boldsymbol{W})y_\alpha = \sum_{a=1}^{N} \eta_{\alpha a}(\boldsymbol{I} - \boldsymbol{D}^{-1}\boldsymbol{W})\phi_a = \sum_{a=1}^{N} \eta_{\alpha a}\lambda_a\phi_a$$

If $\lambda_a$ are in the interval $[\lambda_{\min}, \lambda_{\max}]$, then the following estimate on the norm of $||\hat{y} - y||_D^2$ can be derived:

$$||\hat{y} - y||_D^2 = \sum_{\alpha=1}^{m}\sum_{a=1}^{N}\eta_{\alpha a}^2\lambda_a^2 \leq \sum_{\alpha=1}^{m}\sum_{a=1}^{N}\eta_{\alpha a}^2\lambda_{\max}^2 = ||y||_D^2\lambda_{\max}^2$$

Finally,

$$||\hat{y} - y||_D \leq \lambda_{\max}||y||_D$$

The lower bound on $||\hat{y} - y||$ can be derived similarly:

$$||\hat{y} - y||_D^2 = \sum_{\alpha=1}^{m}\sum_{a=1}^{N}\eta_{\alpha a}^2\lambda_a^2 \geq \sum_{\alpha=1}^{m}\sum_{a=1}^{N}\eta_{\alpha a}^2\lambda_{\min}^2 = ||y||_D^2\lambda_{\min}^2$$

Providing

$$||\hat{y} - y||_D \geq \lambda_{\min}||y||_D$$

### A.2    PROOF OF THE THEOREM 2.2

In this subsection we provide the example sheaf matrices such that vertex feature vectors are mapped to a constant vector.

If $V$ is the set of vertices of the graph $G$ and there is a vector-valued functino $x(v)$ that assigns $n$-dimensional vector of the unit length to each vertex then it simple to show, that there is a vector $\xi$ with a unit length that is not parallel to any of $x(v)$.

The angle between $\xi$ and $x(v)$ is a well-defined continuous function of $x$:

$$\theta(x) = \arccos(\langle \xi, x \rangle)$$

Similarly, the unit vector $\nu(\boldsymbol{x})$ that lies in the plane spanned by $\xi$ and $x(v)$ and is orthogonal to $\xi$ can be introduced:

$$\nu(\boldsymbol{x}) = \frac{\boldsymbol{x} - \langle \xi, \boldsymbol{x} \rangle\xi}{\langle \boldsymbol{x}, \boldsymbol{x} \rangle - \langle \xi, \boldsymbol{x} \rangle^2}$$

Again, because of the assumptions about $\xi$ and $x(v)$, $\nu(\boldsymbol{x})$ is continuous function of $\boldsymbol{x}$.

Therefore, matrix $A(v)$ can be constructed as a composition of the rotation $R_x$ and projection $P_\xi$. The rotation can be wriiten precisely:

$$R_x(f) = \left( f - \langle \xi, f \rangle\xi - \langle \nu(\boldsymbol{x}), f \rangle\nu(\boldsymbol{x}) \right) +$$

$$+ \left( \cos(\theta(\boldsymbol{x}))\xi\xi^\top + \sin(\theta(\boldsymbol{x}))\xi\nu(x)^\top - \sin(\theta(\boldsymbol{x}))\nu(\boldsymbol{x})\xi^\top + \cos(\theta(x))\nu(\boldsymbol{x})\nu(\boldsymbol{x})^\top \right)f$$

The first term is the projection of the vector $f$ to the space of vectors orthogonal to both $\xi$ and $\boldsymbol{x}$, the second term is the rotation in the two-dimensional plane spanned by $\xi$ and $\boldsymbol{x}$. That rotation maps $x$ to $\xi$.

The rotation $\boldsymbol{R}_x$ can be combined with the projection $\boldsymbol{P}_\xi$. The projection $\boldsymbol{P}_\xi$ can be arbitrary: one has to select $m-1$ orthogonal unit vectors that are orthogonal to each other. Therefore, $P_\xi(\xi) = \xi$. Finally, the matrix $A$ is simply:

$$\boldsymbol{A}\big(\boldsymbol{x}(u)\big) = \boldsymbol{P}_\xi\boldsymbol{R}_{\boldsymbol{x}(u)}$$

It is simple to check that the matrix $A$ satisfies orthogonality constraint as in the equation 7. Moreover, vector $x(u)$ is mapped to $\xi$ for any vertex $u$:

$$\boldsymbol{A}(x) = \boldsymbol{P}_\xi\big(\boldsymbol{A}_{\boldsymbol{x}}(\boldsymbol{x})\big) = \boldsymbol{P}_\xi(\xi) = \xi$$

In other words, vertex vector $\boldsymbol{x}$ is lifted to the edge vector $\xi$ for each vertex. Therefore, there is no diffusion in the space of the edge vectors in this case.

## A.3 PROOF OF THE THEOREM 2.3

In the present subsetion we consider the regression problem on a graph $G$ with vertices $V$, We consider the functiob $f : V \to \mathbb{R}$ and feature vectors $\boldsymbol{x}(u)$ that are specified for each vertex $u \in V$. The objective is to learn matrices of sheaf linear transformations $\boldsymbol{A}\big(\boldsymbol{x}(u()$ and approximate $f$ as a linear combination of edge feature vectors:

$$f(u) \approx \zeta^\top \boldsymbol{A}\big(\boldsymbol{x}(u)\big)\boldsymbol{x}(u) = \zeta^\top \boldsymbol{y}(u)$$

Here $\zeta = [\zeta_1, ..., \zeta_m]$ is the vector of coefficients.

In the present section we provide the upper bound for the solution of the regression problem. The ideas is quite simple. If we suppose that first $m$ eigen-functions $\phi_1(u), ..., \phi_m(u)$ of the Hraph-Laplacian operator are in the span of coordinate functions $x_1(u), ...x_n(u)$, then coordinate functions can be projected in the space spanned by $\phi_1(u), ..., \phi_m(u)$. Therefore, there exists such matrix $\boldsymbol{H}$ that for $\beta = 1, ...m$ the following holds:

$$\phi_\beta(u) = \sum_{\alpha=1}^{n} H_{\beta\alpha} x_\alpha(u)$$

Matrix $\boldsymbol{H}$ can be represented as a product of two rotations and diagonal matrix:

$$\boldsymbol{H} = \boldsymbol{R\Lambda U}$$

Here $R, U$ are rotation matrices and $\Lambda$ is a diagonal matrix. We can take first $m$ rows of matrix $\boldsymbol{U}$ as a matrix $\boldsymbol{A}$. It is simple to check that constraints provided by the equation 7 and the equation ?? are satisfied. Moreover, coordinates of the vector $\boldsymbol{y} = \boldsymbol{Ax}$ span the linear space $\phi_1, ..., \phi_m$.

Eigen functions of the Graph-Laplacian operator form orthogonal basis. Therefore, the function $f$ that we would like to approximate can be decomposed as a linear combination of eigen-functions:

$$f(u) = \sum_{a=1}^{N} \xi_a \phi_a(u)$$

Therefore, coefficient $\zeta$ can be adjusted in such a way that

$$\zeta_\alpha^m y_\alpha = p_m(f)$$

Here $p_m$ is the projection on the space spanned by the functions $\phi_1, ..., \phi_m$. Therefore, the mean-squared error is simply the $\ell_2$ norm:

$$\text{MSE} = ||f - p_m(f)||^2$$

Therefore, this is the upper bound of the error that can be achieved during training.

