# OpenReview forum: "Diffusion Minimization and Sheaf Neural Networks for Recommender Systems"
_ICLR.cc/2025/Conference — Submitted to ICLR 2025_

### Official Review · Reviewer_Pi2P · 2024-11-02

**Soundness:** 1
**Presentation:** 1
**Contribution:** 1
**Rating:** 3
**Confidence:** 4

**Summary:**

In this paper, the authors focus on the oversmoothing problem of Graph Neural Networks (GNN), and propose to use Sheaf Neural Networks (SNN) to address the issue of oversmoothing. Specifically, they train SNN to learn user and item embeddings for recommendation via minimization of the classical BPR loss and sheaf diffusion on graphs subjected to orthogonality and consistency constraints. The performance of the proposed technique is evaluated on synthetic test cases and standard benchmarks for recommendations.

**Strengths:**

S1. Solving the oversmoothing problem is significant to GNNs.

S2. The authors impose the consitency constraint for the proposed SheafGCN.

**Weaknesses:**

W1. The novelty of this paper is limited. According to the authors, the idea of combining SNN with recommendation has already been proposed by Bamberger et al. (2024), and the model design of SNN, the orthogonality constraint, and sheaf diffusion were also introduced by Barbero et al. (2022). Therefore, the contribution of this paper is poor.

W2. Some details of the proposed method are not clearly explained, such as:
- The relationship between the expression of the orthogonality constraint in Eq. (7) and its assumption "the dimension of the edge space is (significantly) lower than the dimension of the linear space associated with the vertex" is not obvious and needs to be explained or derived.
- There is no derivation process or rationale provided for obtaining Eq. (12) from Eq. (8), (9), and (10).
- The motivation for minimizing sheaf diffusion is unconvincing. One of the advantages of GNNs is their ability to aggregate information from neighbors to complement the information of individual nodes. However, the authors propose forcing feature vectors not to change in the message-passing procedure, which contradicts the advantage of GNNs.

W3. The experiments in this paper are not convincing:
- The baselines used in the paper include only two classical methods, which do not represent the latest advancements in the field. SheafGCN should be compared with state-of-the-art works such as Barbero et al. (2022) and Bamberger et al. (2024).
- In the recommendation benchmarks, SheafGCN shows a significant gap compared to UltraGCN in metrics like P@10 and R@10, making it difficult to prove the effectiveness of the proposed method.
- The motivation of the paper is to address the oversmoothing problem in GNNs, yet no related experiments have been conducted.

W4. There are no time and space complexity analysis of the proposed method. SheafGCN maintains a learnable mapping function for each node and edge, which results in high training and inference costs on graphs with large-scale nodes and edges. This is impractical and difficult to apply in real recommendation systems.

W5. According to the paper, the propagation of SheafGCN is only related to the feature vectors of nodes. However, according to the authors, these node feature vectors are obtained by training with a recommendation loss, which makes SheafGCN unsuitable for cold-start scenarios since cold-start nodes do not have trained feature vectors.

W6. There are many representation errors in this paper.
- There are many broken references in the text, such as "... reduce the oversmoothing effect ?", "... in the equation ??", "... to the Theorem ??"".
- There are numerous typos and grammatical errors, for example, "... to Graph Neural Networks can.", "... one the graph ..." should be "... on the graph ...".
- In Line 42, "oversmooting" should be corrected to "oversmoothing".
- In Line 169, "We minimizing sheaf diffusion by" should be "We minimize sheaf diffusion by".

**Questions:**

Please refer to the weaknesses in Review.

---

### Official Review · Reviewer_GsEh · 2024-11-04

**Soundness:** 1
**Presentation:** 1
**Contribution:** 2
**Rating:** 3
**Confidence:** 4

**Summary:**

This paper proposes an approach to mitigate the oversmoothing problem in Graph Neural Networks (GNNs) through the use of Sheaf Neural Networks (SNNs). This approach leverages sheaf theory to introduce orthogonality and consistency constraints on the feature diffusion process, minimizing the information loss typically associated with oversmoothing. The methodology is evaluated on 3 datasets, showing results compared to traditional GNN-based recommenders. Key contributions include the introduction of a sheaf-based architecture for recommender systems and a loss function that incorporates orthogonality and consistency constraints, as well as Bayesian Personalized Ranking (BPR) for optimization.

**Strengths:**

1. **Application of Sheaf Theory**:

   The paper introduces the use of sheaf theory in recommender systems, which is an approach to addressing the oversmoothing issue in Graph Neural Networks (GNNs). This could inspire further research in applying advanced mathematical theories to tackle common issues in GNNs and enhance recommender system performance.

2. **Theoretical Foundation and Formulation**:

   The authors provide a rigorous theoretical foundation, including the formulation of orthogonality and consistency constraints for minimizing diffusion in feature propagation.

**Weaknesses:**

1. **Numerous Presentation Issues**:

   The presentation of the paper is marred by numerous issues. There are several confusing expressions, such as the use of parentheses in "linear transformations A(x(u()" on line 652, which could mislead readers. Citation inconsistencies, such as the placeholder "Theorem ??" on line 315, further reduce clarity and professionalism. Additionally, there are formatting errors, like missing borders in Tables 1 and 2, which affect the readability of the results. Overall, these presentation issues significantly detract from the paper’s quality.

2. **Baseline Selection**:

   The paper only compares the proposed method to two outdated models (LightGCN [1] and UltraGCN [2]), limiting the relevance and validity of its experimental comparisons.

3. **Lack of Experimental Details**:

   The experimental setup lacks transparency, making it difficult to reproduce results or assess the robustness of the comparisons.

4. **Lack of Complexity Analysis:**

   The method proposed in this paper lacks an analysis of time complexity, raising concerns about its efficiency.

[1] He, Xiangnan, et al. "Lightgcn: Simplifying and powering graph convolution network for recommendation." *Proceedings of the 43rd International ACM SIGIR conference on research and development in Information Retrieval*. 2020.

[2] Mao, Kelong, et al. "UltraGCN: ultra simplification of graph convolutional networks for recommendation." *Proceedings of the 30th ACM international conference on information & knowledge management*. 2021.

**Questions:**

1. Why is there a significant difference in line thickness between the two images used in Figure 1? Is there a particular reason for this?

2. In the experimental data presented in Table 1, **why is UltraGCN's R@10 higher than UltraGCN's R@20?** Is it because the authors are using a different definition of Recall than commonly used, or is it a mistake in the writing?

---

### Official Review · Reviewer_iNMZ · 2024-11-05

**Soundness:** 2
**Presentation:** 2
**Contribution:** 2
**Rating:** 3
**Confidence:** 3

**Summary:**

The paper proposes a novel approach to train Sheaf Neural Networks (SNNs) for recommender systems by jointly minimizing the BPR loss, sheaf diffusion, and orthogonality/consistency constraints on the sheaf parameters. Experiments on synthetic and benchmark datasets demonstrate the effectiveness of the proposed technique.

**Strengths:**

1. The paper addresses the important issue of over-smoothing in Graph Neural Networks (GNNs), which can adversely affect their performance in recommender systems.

2. The proposed approach is theoretically grounded and includes a mathematical formulation and analysis of the constraints and loss functions.

3. The authors have conducted experiments on synthetic datasets and standard recommendation benchmarks to validate the effectiveness of their technique

**Weaknesses:**

1. The paper has numerous formatting issues, such as missing table lines (e.g., Table 1, Table 2), and incorrect citation formats (e.g., should use \citep{}, ? in line 045), which need to be addressed.

2. The writing could be improved to highlight better the novelty and contributions of the proposed method specifically for recommender systems. As the authors mentioned, Sheaf Neural Networks and the use of constraints (Section 2.2) are not entirely new concepts.

3. While the paper claims to address the oversmoothing problem, it lacks an in-depth experimental analysis of the oversmoothing issue, such as studying the performance degradation with increasing layer depth in GNNs. For example, compare the performance of proposed methods versus baseline GNNs and other methods mentioned in the following survey [1] that tackle the over-smoothing problem as the number of layers increases.

[1] A Survey on Oversmoothing in Graph Neural Networks. 2023.

**Questions:**

1. What is the specific motivation for tailoring this method to recommender systems? The methodology seems to be generally applicable to Sheaf Neural Networks, regardless of the application domain.

2. How does the proposed approach compare to other existing techniques for mitigating over-smoothing in GNNs, such as regularization, stochastic edge dropout, or residual connections?

---

### Official Review · Reviewer_TqYC · 2024-11-05

**Soundness:** 2
**Presentation:** 2
**Contribution:** 3
**Rating:** 3
**Confidence:** 5

**Summary:**

This paper studies the problem of over-smoothing in Graph Neural Networks, particularly in the context of recommender systems. The main contribution of the paper is a novel training framework for Sheaf Neural Networks that incorporates orthogonality, consistency constraints, and diffusion minimization, along with a hierarchical optimization of loss functions to preserve unique node features and improve recommendation quality.

**Strengths:**

S1. The paper extends Sheaf Neural Networks specifically to recommender systems, and experimental results show its effectiveness.

S2. I like the idea of hierarchical optimization of loss functions, as it prioritizes different constraints to enhance model robustness.

**Weaknesses:**

W1. The motivation for addressing over-smoothing in this paper is unclear, as it lacks a thorough explanation of its specific impact on recommender systems and why existing methods are insufficient.

W2. The baseline algorithms compared in the paper are fairly old. I would suggest the authors to compare SheafGCN to some more recent work, for example [1,2,3]. Additionally, the performance of LightGCN presented by the authors is significantly worse than the results reported in [3]. Please explain the reason for this discrepancy.

W3. The presentation of the paper requires improvement due to numerous typos, unclear explanations, and inconsistent formatting, which detract from its professionalism and readability.


[1] Guo, Jiayan, Lun Du, Xu Chen, Xiaojun Ma, Qiang Fu, Shi Han, Dongmei Zhang, and Yan Zhang. On Manipulating Signals of User-Item Graph: A Jacobi Polynomial-based Graph Collaborative Filtering. In KDD, 2023.

[2] Yu, Junliang, Hongzhi Yin, Xin Xia, Tong Chen, Lizhen Cui, and Quoc Viet Hung Nguyen. Are graph augmentations necessary? simple graph contrastive learning for recommendation. In SIGIR, 2022.

[3] Zihan Lin, Changxin Tian, Yupeng Hou, Wayne Xin Zhao. Improving Graph Collaborative Filtering with Neighborhood-enriched Contrastive Learning. In WWW, 2022.

**Questions:**

see W1-W3.

---

### Meta-Review · Area_Chair_WaFM · 2024-12-21

**Metareview:**

The paper proposes a new approach with Sheaf Neural Networks to address the important issue of over-smoothing in GNNs for recommender systems. The problem is important. Experiments show that the proposed model outperforms baseline methods. Some theoretical foundations are provided. Many issues need to be addressed, such as insufficient baseline comparison and related work discussion, unclear motivation, lack of experimental details/complexity analysis, and writing problems. Reviewers are negative about this work.

**Additional Comments On Reviewer Discussion:**

No discussion is necessary as all give rejection.

---

### Decision · Program_Chairs · 2025-01-22

Reject